# AC amplification gain in organic electrochemical transistors for impedance-based single cell sensors

Filippo Bonafè [1], Francesco Decataldo [1], Isabella Zironi [1], Daniel Remondini [1], Tobias Cramer [1] ✉ & Beatrice Fraboni [1]

Research on electrolyte-gated and organic electrochemical transistor (OECT) architectures is motivated by the prospect of a highly biocompatible interface capable of amplifying bioelectronic signals at the site of detection. Despite many demonstrations in these directions, a quantitative model for OECTs as impedance biosensors is still lacking. We overcome this issue by introducing a model experiment where we simulate the detection of a single cell by the impedance sensing of a dielectric microparticle. The highly reproducible experiment allows us to study the impact of transistor geometry and operation conditions on device sensitivity. With the data we rationalize a mathematical model that provides clear guidelines for the optimization of OECTs as single cell sensors, and we verify the quantitative predictions in an in-vitro experiment. In the optimized geometry, the OECT-based impedance sensor allows to record single cell adhesion and detachment transients, showing a maximum gain of 20.2±0.9 dB with respect to a single electrode-based impedance sensor.

In recent years, electrolyte gated and electrochemical transistor architectures have found broad attention in biosensor research[1,2]. The interest is motivated by the possibility to combine electrochemical transduction mechanisms known from metallic electrodes with the intrinsic amplification properties of a transistor structure[3]. The amplification is highly wanted to improve signal-to-noise ratio in challenging sensor applications aiming for instance at biochemical detection in complex mixtures[4] or single-cell bioelectronic monitoring[5]. Significant progress on this concept has been made by the introduction of organic or carbon-based semiconducting materials with high stability in water such as carbon nanotubes[6,7], graphene[8,9], or organic semiconductors[10]. When patterned into a semiconducting channel connected to a source and drain electrode, these materials enable transistor-like structures to replace single metallic working electrodes. Since the semiconducting channel is in direct contact with the aqueous electrolyte, a strong capacitive coupling results between the channel's electronic carrier concentration and the electrochemical potential in the solution[11]. Events that alter the electrochemical potential and impact on the ionic

distribution at the channel–electrolyte interface thus gate the sensor's semiconducting channel conductivity. Even small perturbations in the ionic distribution can cause a large variation in the number of electronic carriers flowing through the channel from source to drain electrode, hence leading to an amplification effect[12]. This qualitative argument is demonstrated by improved sensitivity in several applications realized with electrolyte gated transistor structures[13]. However, to achieve optimized sensor devices, a quantitative understanding of such signal gain is needed. The goal of this work is to derive a quantitative model that relates transistor amplification gain to semiconductor material properties and electrolyte gated transistor architecture.

A widely explored class of electrolyte gated transistors are organic electrochemical transistors (OECTs)[14]. OECTs exploit organic mixed ionic and electronic conductors such as poly(3,4-ethylenedioxythiophene):polystyrene sulfonate (PEDOT:PSS)[2]. The high chemical stability, the facile, solvent-based processing and the high biocompatibility make PEDOT:PSS a material of choice for healthcare applications[15]. In PEDOT:PSS, electronic transport is achieved by

[1]Department of Physics and Astronomy, University of Bologna, Viale Berti Pichat 6/2, 40127 Bologna, Italy. ✉e-mail: tobias.cramer@unibo.it

mobile hole charges present in the oxidized semiconducting polymer PEDOT. The positive charge of the holes is counterbalanced by fixed, negative ionic charges of the polyanion PSS[16]. Oxidized organic semiconductor and ionic polyanion form a nanophase separated network that generates close electrostatic interaction between the two phases combined with efficient electronic as well as ionic charge transport[15,17]. The use of secondary dopants such as dimethyl sulfoxide (DMSO) and ethylene-glycol (EG) leads to further separation of PSS-rich islands from the conductive network of PEDOT, and therefore to a better conduction pathway and an increase of electrical conductivity[18]. Consequently, high electronic carrier mobility ($\mu > 10$ cm$^2$/V/s)[19] is combined with a strong volumetric capacitive coupling between the ionic phase and the electronic phase ($c > 30$ F/cm$^3$)[20]. In OECTs, these properties' combination gives rise to large values of transconductance $g_m = \partial I_{ch}/\partial V_g$ that express how small changes in the electrostatic potential $\partial V_g$ of the electrolyte gate the electronic channel current $I_{ch}$[21]. Relying on the large transconductance combined with the biocompatible material properties, many research works propose OECTs as amplifying transistor to be integrated in electrochemical and bioelectronic sensors for healthcare applications[22]. Established examples regard biosensors used to quantify the concentration of ionic or redox active analytes[23]. In such devices, selectivity towards specific analytes such as DNA or RNA biomarkers[24], enzymes[25], or immunoglobulins was demonstrated by device functionalization with biorecognition elements[26]. Other successful applications of OECTs regard their use as potentiometric sensors for electrophysiological signals[27]. In this case, the electrical circuit involving the OECT must operate ultimately as a voltage amplifier, that produces an amplified output voltage ready for digitization. Gain is then expressed as the ratio between the original potentiometric signal and the output voltage. Depending on the circuit design, DC amplification gains reaching 30 V/V have been demonstrated in OECT based potentiometric sensors[28].

A second, emerging class of biosensors that takes advantage of OECT amplification regards impedance based sensors for monitoring cellular adhesion and cell layer barrier properties as quantified by the transepithelial electrical resistance (TEER)[29]. Cell adhesion is an essential process in cell communication and regulation and becomes of fundamental importance in the development and maintenance of tissues[30]. Changes in cell adhesion can be the defining event in a wide range of diseases, including arthritis[31], cancer[32], osteoporosis[33], and atherosclerosis[34]. The study of single-cell adhesion is one of the most important and complicated aspects to understand in life sciences, with a considerable potential impact in bioelectronics. Over the years, numerous studies have shown the use of different techniques for the analysis of single-cell adhesion. Both the traction force microscopy (TFM)[35] and micropillar-array technique measure the cell adhesion force by monitoring the deformation induced on an elastic substrate[36]. Other methods include the use of atomic force microscopy[37] and optical tweezers[38] of microfluidics to assess the impact of cellular shape, size, and deformability during adhesion[39]. Despite their success in different demonstrations, these techniques rely on expensive equipment, are typically time consuming and potentially can alter the cell behavior[40]. An alternative non-invasive approach that combines scalability and real-time monitoring, is offered by electrical measurements that probe the electric cell–substrate impedance[41]. In this technique, the cells are allowed to adhere directly onto the conductive surface of a sensor. A small AC voltage is applied and the ionic current that passes through the layer of adhering cells to the sensor is measured[42]. Changes in cellular adhesion or intercellular barrier properties change the measured current signal thus providing a simple means for real-time monitoring[43]. By replacing the metallic working electrode of a traditional impedance sensor with a electrolyte gated OECT, amplification of the current signal is achieved thus increasing sensitivity and reducing possible noise pickup[44]. Current amplification becomes particularly important in high impedance applications as

encountered when the sensor size is scaled down to micrometric scales matching cellular dimensions. Such downscaling opens the opportunity to translate impedance based cellular monitoring to the single-cell level[45]. This ultimate sensor resolution is highly desired in biomedical research as the importance of single-cell phenotyping is increasingly recognized for the study of cell development and physiology, as well as for research on cellular pathologies such as cancer[46]. An important step in this direction was recently achieved by Hempel et al. by demonstrating single-cell sensitivity in an OECT enabled impedance sensor. The authors found that significant differences in the sensors transfer function are caused by cells adhering to the transistor surface. Such changes in transfer function are usually quantified as the transistor bandwidth (or response time) and equivalent circuit models were developed to explain the observed frequency response and its relation to impedances at the cell/PEDOT:PSS interface[5].

Despite the promising results, a quantitative study on the transistor amplification gain in the frequency spectrum relevant for cellular impedance sensing is still lacking. Different studies demonstrate that the high OECT transconductance is limited to the low frequency regime and strongly depends on transistor geometry and materials properties[21]. Consequently, there is a need for a quantitative model that relates impedance sensor gain to OECT transconductance and other device properties in order to enable a rational optimization of OECT based impedance sensors. A clear understanding of gain is also desired to define when OECT amplification has significant advantages over one-terminal, low-impedance microelectrode-based sensors offering simpler fabrication and electrical operation. To overcome the issue, we introduce in this work a model experiment that allows a quantitative analysis of amplification gain in OECT based impedance sensors. As cellular in vitro experiments are inherently difficult to control we substitute the cell by a dielectric microparticle of similar dimensions. We control the position of the microparticle on top of the microscale impedance sensors with an AFM and achieve highly reproducible measurements that enable to compare the current output of OECT based sensors with equivalent microelectrode sensors. To rationalize the findings, we develop an analytical model that describes the gain as a function of the applied frequency, the device geometry and PEDOT:PSS materials properties. Relying on this model, we design an optimized device and demonstrate its efficiency by measuring the transients of single-cell adhesion and detachment in in vitro experiments. Noteworthy, we observe a significant AC gain reaching values of ($20.2 \pm 0.9$) dB for the transistor structure, thereby demonstrating the advantages arising from the OECT amplification in single-cell impedance sensing experiments.

## Results

### Impedance sensing of a dielectric microparticle

The first objective of our work is to introduce a novel experiment to quantify the sensitivity of electrochemical impedance sensors operated in OECT or microelectrode configuration. To this end we realized the experimental setup shown in Fig. 1a, b. The set-up contains a dielectric microparticle (with diameter of 50 μm) attached to the bottom part of an AFM cantilever to have micrometric control of its position in the 3 spatial directions (Fig. 1c). Once the microparticle is finely aligned with the x-y coordinates on the center of the sensor surface, we use the z-stage of the microscope to control the particle-sample distance d. Contact of the microparticle with the sensor surface is determined by the onset of a repulsive force acting on the AFM cantilever. The electric circuit to operate the OECT impedance sensor contains an Ag/AgCl wire that is used as the gate electrode, controlling the electrical potential of the aqueous electrolyte solution (0.1 M PBS). A DC voltage $V_{D,DC}$ is applied between the source (S) and drain (D) electrodes of the OECT to drive the electronic current $I_{D,DC}$ in the PEDOT:PSS channel. The measured transfer and output characteristics

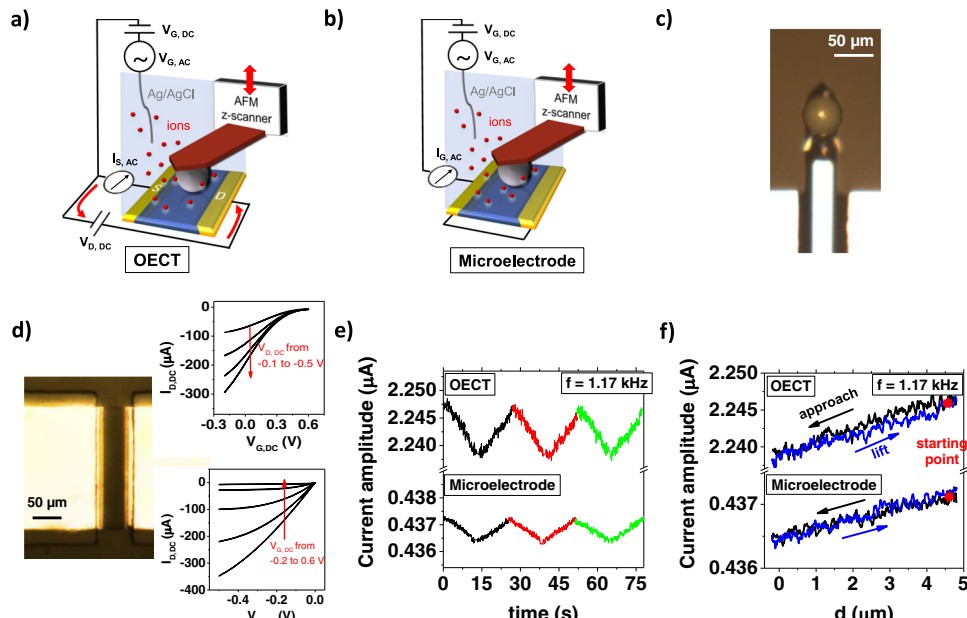

**Fig. 1 | Impedance sensing of a dielectric microparticle. a, b** Schematic of the experimental set-up in OECT and microelectrode configuration. **c** Optical microscopy image of the microparticle attached to the AFM cantilever. **d** Optical image of the active PEDOT:PSS channel ($W \times L = 200 \times 50\,\mu m$) in an OECT device, whose DC transfer and output characteristics using a Ag/AgCl gate are shown in the inset. **e** Variation of the AC current amplitude in OECT and microelectrode configuration during repeated microparticle approach and retract. Consecutive measurements are indicated with different colors. **f** Detail on a single microparticle distance−AC current measurement acquired with an OECT and a microelectrode device. Different colors are used to indicate the microparticle approach and lift. The slope of the curves yields the sensitivity $s$ of the impedance sensor.

of a typical OECT (Fig. 1d), demonstrate that the gate voltage effectively modulates the channel current. (The entire set of transfer curves of the devices with different channel dimensions used in the experiments is reported in Supplementary Fig. 1). For impedance sensing, we superimpose a small sinusoidal oscillation signal $V_{G,AC}$ (with amplitude 10 mV and angular frequency $\omega$) on the gate bias $V_{G,DC}$. This leads to an AC current in the PEDOT:PSS layer[47], which is measured by a lock-in amplifier connected to the source contact ($I_{S,AC}$).

In the microelectrode configuration (Fig. 1b), the circuit is simplified, as source and drain electrodes are in short circuit and are jointly connected to the lock-in amplifier. Therefore, no OECT channel current is present, and all the electric current measured during impedance sensing is the gate current $I_{G,AC}$, flowing from the electrolyte into the PEDOT:PSS layer. All other components are identical to the OECT configuration to permit a direct comparison here conducted at 5 different frequencies (0.12, 0.33, 1.17, 3.33 and 11.7 kHz).

In Fig. 1e, we show the results of a typical microparticle distance−AC current experiment conducted at 1.17 kHz excitation frequency, which was chosen as an example. The amplitudes of the AC currents in OECT and microelectrode configuration are plotted as a function of time. During the experiment, the microparticle is approached and retracted from the device channel for three consecutive times. For both configurations, the current amplitude follows the motion of the microparticle in a highly reproducible manner over consecutive cycles, highlighting the stability of the characterization method. In Fig. 1f, the same data are plotted as a function of the distance between microparticle and sensor surface (normalized data are reported in Supplementary Fig. 4). Both types of devices produce a reversible, linear response in which the approach leads to a reduction in AC amplitude. Qualitatively, this response is expected, as the microsphere represents a barrier for the ionic current in the electrolyte: when it is close to the sensor surface, the half space through which ions can approach the active layer is reduced, thus increasing the effective impedance of the electrolyte $Z_{el}$. Consequently, upon approach, the interfacial impedance measured with the sensor increases and the AC current amplitude drops. We note that in first order approximation a similar response is expected when a biological cell adheres to the sensor surface.

The results are crucial for our goal as they permit the quantitative assessment of the sensitivity of the impedance sensor. For the case of a high sensitivity, small changes in the impedance $Z_{el}$ cause large variation in AC current amplitude. Therefore, we define the sensitivity as $s = \partial I_{AC}/\partial Z_{el}$. In our experiment, $\partial Z_{el}$ is directly related to the microparticle displacement $\partial Z_{el} = p \cdot \partial d$. The proportionality constant $p$ quantifies how much the electrolyte impedance changes when the microparticle-channel distance $d$ is modified. Therefore, $p$ is independent on the sensor configuration (OECT vs microelectrode) but is only a measure of the variation of the electrolyte impedance when the geometry of the ionic current barrier is modified. We obtain its numerical value for each channel geometry by fitting the microelectrode impedance spectrum (see Supplementary Fig. 5). Accordingly, the sensitivity is given by the slope of the approach curves shown in Fig. 1f. For the channel geometry of $W \times L = 100 \times 100\,\mu m$, we obtain values of $s_{OECT} = 0.059 \pm 0.002\,nA/\Omega$ and $s_{\mu E} = 0.023 \pm 0.006\,nA/\Omega$. The values show a greater sensitivity in the OECT device with respect to the microelectrode, due to the contribution of OECT channel current to the AC response.

## Quantitative model for PEDOT:PSS-based impedance sensors

We developed an analytical model to express the impedance sensitivity $s$ as a function of the sensor operation conditions, material properties and geometry. Objective is a quantitative understanding of the factors that increase sensitivity in OECT configuration with respect to PEDOT:PSS microelectrodes. Many studies decouple charge transport in OECTs in an electronic and an ionic circuit[14]. A schematic of this representation is reported in Fig. 2a, where the components of the electronic and the ionic circuit are indicated in blue and orange, respectively. Electronic charge carriers (holes) are driven by the drain voltage $V_{D,DC}$ and carry the channel current in an OECT, while ionic charge carriers are driven by the gate voltage

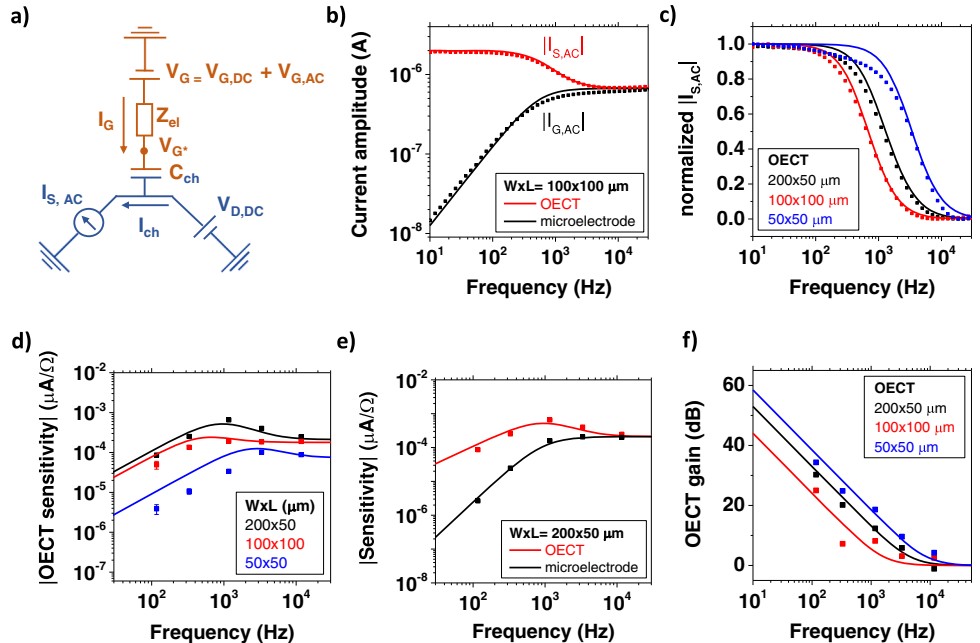

**Fig. 2 | Quantitative model for PEDOT:PSS-based impedance sensors.**
**a** Equivalent circuit of an OECT during impedance sensing. Orange and blue colors indicate ionic and electronic parts of the circuit, respectively. **b** Modeling of the current frequency spectra of an OECT and a PEDOT:PSS microelectrode. **c** Normalized OECT source current amplitudes as a function of the applied frequency for different channel geometries. **d** OECT sensitivity during the microparticle impedance sensing experiment. The error bars are obtained by averaging between the approach and lift curves of the AFM experiment. **e** Comparison between the sensitivity of an OECT and a microelectrode having the same dimensions. **f** OECT gain for different channel geometries.

$V_G = V_{G,DC} + V_{G,AC}$ and modulate the concentration of holes and, consequently, the electronic conductivity of the transistor channel. The limited conductivity of the electrolyte as well as the presence of dielectric objects close to the sensor surface generate an impedance $Z_{el}$, which causes a potential drop in the electrolyte, and the voltage at the electrolyte/channel interface $V_{G^*,AC}$ that acts on the channel and determines the drain current[42]. For the impedance sensing, we are interested in the AC response of the transistor and we express the AC current flowing in the OECT channel as $I_{ch,AC} = g_m \cdot V_{G^*,AC}$. Following Bernards model[48], $g_m$ can be expressed as $g_m^{lin} = -\frac{W}{L}\mu c_v t V_{D,DC}$ and $g_m^{sat} = -\frac{W}{L}\mu c_v t (V_{G,DC} - V_t)$ in linear or saturation conditions. In these expressions, $W$, $L$, and $t$ indicate the width, length and thickness of the sensor channel, $\mu_p$ the holes mobility, $c_v$ the volumetric capacitance of PEDOT:PSS and $V_t$ the OECT threshold voltage.

To derive the overall AC current, it is important to note that in AC transport conditions the source (and drain) current signals are composed of two contributions:

$$I_{S,AC} = I_{ch,AC} + f_{OECT} \cdot I_{G,AC} \qquad (1)$$

The first ($I_{ch,AC}$) originates from the channel current, whereas the second ($I_{G,AC}$) is due to the gate current and regards the capacitive current that has increasing importance at higher frequencies. Its value is given by $I_{G,AC} = V_{G,AC}/Z_G$ in which $Z_G = Z_{el} + Z_{ch}$ is the overall impedance of the sensor given by the series combination of the electrolyte impedance $Z_{el}$ and the impedance related to the PEDOT:PSS channel capacitance $Z_{ch} = 1/(i\omega C_{ch})$. The channel capacitance can further be related to the geometry and the volumetric capacitance of the PEDOT:PSS layer. Possible contributions due to parasitic capacitances are neglected for simplicity.

The factor $f_{OECT}$ in Eq. (1) determines how the gate current is distributed between the source and the drain terminal[49]. In general, the factor $f_{OECT}$ depends on the bias conditions ($V_{D,DC}$ and $V_{G,DC}$), on channel geometry and on AC or DC measurement conditions[50]. In our case, we consider the AC transport regime where the gate current is a

pure capacitive current without Faradaic contributions. Further, in our biasing conditions ($V_{G,DC} = 0.1$ V and $V_{D,DC} = -0.4$ V) a significant negative potential is applied to the drain electrode leading to a depletion of holes from the channel region nearby the drain contact[17]. As a result, the capacitive gate current encounters a resistive barrier at the drain electrode and instead enters into the source electrode. For this reason, we set $f_{OECT} = 1$ in our data analysis for each sensor geometry. The value is supported by numerical fitting procedures of our frequency-dependent data leading to values close to one (see Supplementary Table 1).

The figure of merit of the OECT impedance sensor (the sensitivity $s_{OECT}$) quantifies its capability to transduce a variation of $Z_{el}$ into a current output. This can be calculated from the model by differentiating Eq. (1):

$$s_{OECT} = \left| \frac{\partial (I_{ch,AC} + f_{OECT} I_{G,AC})}{\partial Z_{el}} \right| = |s_{ch} + f_{OECT} \cdot s_{\mu E}| \qquad (2)$$

After inserting the expressions for the two AC current contributions and differentiation, we obtain for the channel sensitivity

$$s_{ch} = \frac{g_m}{Z_G}\left(1 - \frac{Z_{el}}{Z_G}\right)V_{G,AC} \qquad (3)$$

and the sensitivity of the microelectrode

$$s_{\mu E} = \frac{1}{Z_G^2}V_{G,AC} \qquad (4)$$

The explicit mathematical expressions relating the devices' sensitivity to the applied frequency are reported in Supplementary Discussion 1. The suitability of this simple approach to model the AC response of an OECT is demonstrated in Fig. 2b, c. Figure 2b compares the frequency response of an OECT and of a microelectrode with the model predictions. The PEDOT:PSS channel width and length are $W = 100$ μm and $L = 100$ μm, respectively. The channel capacitance and

the electrolyte resistance $R_{el}$ were extracted for each device geometry by fitting the microelectrode impedance spectrum with an equivalent RC circuit. The average volumetric capacitance of PEDOT:PSS resulted to be $c_v = 28 \pm 2\,F/cm^3$, obtaining a result consistent with literature findings[20]. The OECT device shows a significantly higher current in the low frequency domain. Here the electronic channel current $I_{ch}$ prevails, and the transistor demonstrates clear amplifying properties. Then, above a cutoff frequency $f_c = 625\,Hz$, the transistor response is limited by the slow ionic transport between the channel and the electrolyte[51]. At the same time the microelectrode's response increases with frequency until a current limitation is reached due to the electrolyte impedance. As a consequence, in the high frequency limit both impedance sensor configurations yield the same current response. Importantly, the cutoff frequency that determines the OECT amplification is determined by the channel geometry as demonstrated in Fig. 2c. The plot of the current amplitude versus frequency for OECTs with different channel sizes clearly shows that with increasing channel area and length, a strong reduction in $f_c$ is observed.

Finally, we systematically study the OECTs sensitivity towards electrolyte impedance changes with the microsphere experiment introduced above. Figure 2d shows the measured values for $s_{OECT}$ obtained for three different channel geometries at different AC frequencies. Equation (2) is in excellent agreement with the frequency dependence of the measured data, allowing to resume the main findings of the model with the following statements. (i) In the low frequency regime, the OECT sensitivity shows a linear increase with frequency until it reaches a sensitivity maximum $s_{OECTmax}$. In this frequency range the sensitivity increases strongly with the channel aspect ratio $W/L$ as it is highly controlled by the OECT transconductance. (ii) In the intermediate regime, OECT impedance-based sensors have a maximum sensitivity at a defined operation frequency, which corresponds to their low-pass cutoff $f_c$. The spectral position of $f_c$ (see Supplementary Discussion 2) is mainly determined by the channel area, which defines the channel capacitance $C_{ch}$ and the electrolyte resistance $R_{el}$. Changes in the aspect ratio $W/L$ modify the OECT transconductance and have a direct impact on the value of $s_{OECTmax}$. Accordingly, the frequency cutoffs of the $100 \times 100\,\mu m$ and the $200 \times 50\,\mu m$ structures are almost coincident, but the latter shows a higher sensitivity maximum. The smaller dimensions of the $50 \times 50\,\mu m$ device shift its $f_c$ towards higher frequency, while $s_{OECTmax}$ is limited by the square aspect ratio. (iii) In the high frequency limit, the OECT sensitivity results from the capacitive gate current and is mostly controlled by the area of the channel. This is also evident from Fig. 2e, where we compare the sensitivity of a microelectrode and an OECT with the same dimensions ($W \times L = 200 \times 50\,\mu m$). The transistor amplification, which is significant at low frequencies, has a relevant impact on the sensitivity. However, at high frequencies the response of both devices is limited by the electrolyte resistance and no significant differences are present. Such an observation is reflected by a frequency-dependent OECT gain, which can be directly calculated with our model from Eqs. (2) and (4):

$$\text{gain}_{OECT} = 20 \cdot \log_{10}\left(\left|\frac{s_{OECT}}{s_{\mu E}}\right|\right) = 20 \cdot \log_{10}\left(\frac{g_m}{\omega C_{ch}} + f_{OECT}\right) \quad (5)$$

The OECT gain is highest in the low-frequency regime, but is still significant in the 0.1–10 kHz range, where the impedance of the cell layers is typically measured[52]. This justifies the use of a transistor structure for high-precision bioelectronic impedance sensing experiments[5]. Figure 2f demonstrates that the OECT gain is a geometry-dependent parameter. The smallest device ($W \times L = 50 \times 50\,\mu m$) shows the highest gain, while a rectangular channel geometry is preferable for OECTs with the same area, since the transconductance increases with the $W/L$ ratio while the channel capacitance remains constant.

## Single-cell impedance sensor experiment

We demonstrated the value of the mathematical model here proposed for the optimization of a PEDOT:PSS-based single-cell impedance sensor by monitoring single-cell adhesion and detachment in an in vitro experiment, simultaneously measuring the impedance changes with both an OECT and a microelectrode. According to the model prediction and the AFM experiment, we patterned the device channels with a $200 \times 50\,\mu m$ rectangular geometry, which provides the best performances in terms of sensitivity. The T98G cell line cultured in Minimum Essential Medium was diluted to have a final density of $1 \times 10^3\,cells/cm^3$ and poured on the surface of the impedance sensors (see the "Methods" section for full details). After seeding, the cells reached the underlying substrate by gravity. We microfabricated a linear array of 10 PEDOT:PSS channels (see Supplementary Fig. 6) to largely increase the probability of a single-cell settling onto a sensor. An optical image of the final experimental configuration is reported in Fig. 3a, showing a single T98G cell positioned at the center of the PEDOT:PSS channel. The encapsulation of the metallic electrodes with negative photoresist insulates the device from all the remaining cells which are not lying in the PEDOT:PSS active layer. We acquired the current spectra of both the OECT and the microelectrode at consecutive time intervals to make a real-time detection of the cell adhesion process. To stress the full consistency of the measurements acquired with the OECT and the microelectrode, we plot in Fig. 3b the current amplitudes measured at 625 Hz as a function of time. The sensing frequency was selected in correspondence to the OECT cutoff (measured at time $t = 0$), where the model indicates the maximum sensitivity. The signals acquired at the beginning of the experiment are stable around a maximum value. Afterward, at time $t = 20$ min the current start to decrease, indicating the beginning of the cell adhesion process. This produces a rapidly varying response until $t = 60$ min, when the decrease becomes slower, and the current stabilizes around a minimum value. At $t = 200$ min, we used a cell dissociation agent (trypsin) to completely remove the cell from the sample surface, and devices recovered their original current amplitude.

We report in Fig. 3c the full current spectra acquired before and after the treatment with trypsin ($t = 180$ min and $t = 240$ min, respectively). The cell adhesion produced a large shift of the OECT low-pass cutoff towards smaller frequency. In parallel, the current spectrum of a control device placed in the same reservoir, but with no cell seeded on the PEDOT:PSS layer, remained unaltered (see Supplementary Fig. 7). After trypsinization, the initial cutoff frequency is fully recovered. These combined observations clearly demonstrate that the cutoff shift is only caused by the single-cell adhesion on the PEDOT:PSS layer. The same considerations can be extended to the PEDOT:PSS microelectrode. Here, the cell adhesion process is revealed by a decrease in the gate current amplitude, which reaches its minimum at $t = 180$ min, coherently with the OECT measurements. After the cell detachment ($t = 240$ min), the current increases to its original values.

To provide a direct comparison between the sensing performances, we averaged the current amplitudes acquired when the cell is detached ($t < 30$ min and $t = 240$ min) and attached ($120 < t < 180$ min), and we subtracted the resulting values to calculate the experimental sensitivity for both devices. Repeating this analysis in the whole frequency spectrum, we obtained the curves reported in Fig. 3d, that confirm the results of the quantitative AFM experiments (Fig. 2e). The transistor amplification has a significative impact on the device sensitivity in the frequency range between $10^2$ and $10^4$ Hz, with a peak at 625 Hz, corresponding to the OECT cutoff. On the other hand, when the modulation frequency is high, the OECT transconductance becomes negligible, and the transistor structure does not offer substantial advantages with respect to a microelectrode. The OECT gain (Fig. 3e) was calculated by applying Eq. 5 and reaches a value of $20.2 \pm 0.9$ dB at the highest sensitivity point (625 Hz).

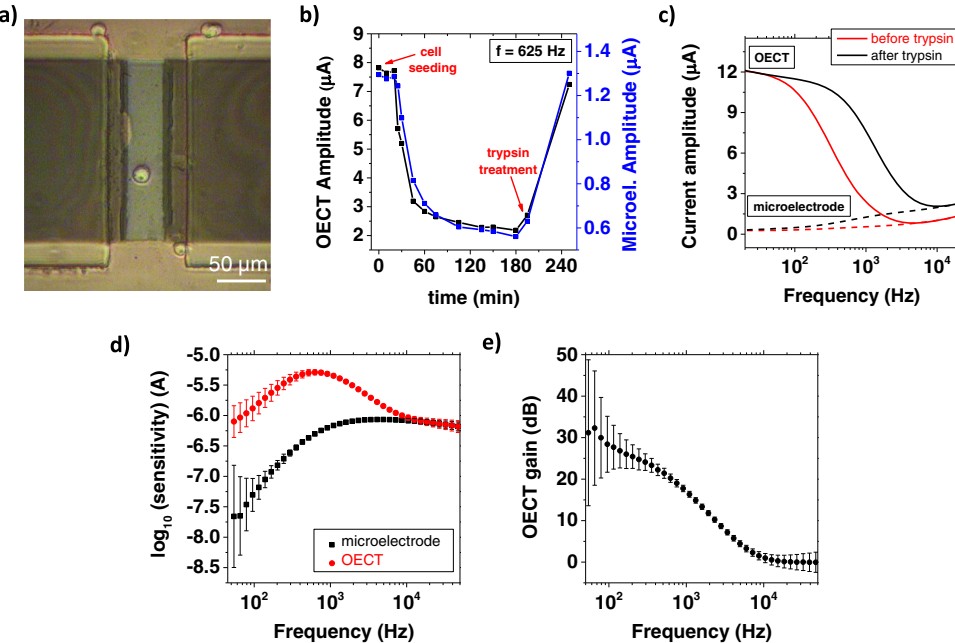

**Fig. 3 | Single-cell impedance sensor experiment. a** Optical image acquired after seeding a single cell at the center of a PEDOT:PSS sensing channel. **b** Time evolution of the cell adhesion monitored with an OECT and a microelectrode. **c** Current spectra acquired in the OECT (straight line) and microelectrode (dashed line) configuration before and after trypsinization. **d** Experimental OECT and microelectrode sensitivity. Error bars are obtained by averaging between $n = 4$ measurements acquired at different time. **e** Experimental OECT gain. Error bars are calculated from the experimental sensitivities by applying Eq. 5.

The frequency response of the OECT gain is well described by our model for both experiments, the single cell as well as the dielectric particle detection. However, the OECT current amplitude reduction is much larger for the case of the single cell even though the cell body has a diameter that is smaller than the dielectric particle. The effect is attributed to the much larger impedance increase caused by the cell adhering to the sensor surface. Glioblastoma tumor cells such as T98G secrete large amounts of laminin and glycoproteins to self-assemble the basement membrane below their cellular body[53]. We hypothesize that the basement membrane spreads below the cell body on top of the PEDOT:PSS channel and acts as a barrier increasing significantly the impedance. Trypsin treatment removes the cell body and dissolves also the basement membrane, making the effect reversible

## Discussion

In this paper, we present a quantitative analysis of the performance of OECTs as single-cell impedance sensors. We introduce a model experiment where we simulate the detection of a single cell by the impedance sensing of a dielectric microparticle. The microparticle is attached to an AFM cantilever to have a refined control of its positioning on the sensor surface. By approaching and retracting the microparticle from the sensor surface, we introduce a controlled impedance change that permits to quantify the sensitivity of the impedance sensor. With the model experiment we can compare in a reproducible manner the sensitivity of different OECT or microelectrode-based sensors. To rationalize the experimentally determined sensitivities, we develop a mathematical model. The model correctly predicts the dependence of the sensitivity on frequency, sensor geometry and semiconductor materials properties such as carrier mobility and volumetric capacitance. From these findings, we derive two major conclusions for OECT based impedance sensors: (i) depending on geometry and semiconductor materials properties, the sensitivity has a maximum at a defined operation frequency; (ii) the OECT based sensor has a significant gain with respect to a simpler microelectrode-based sensor. The gain increases towards lower frequencies due to the increased contribution of the transistor channel

current. Instead, towards very high frequencies the microelectrode approaches the OECT performance as the overall current is dominated by the ionic gate current. Smaller OECT channel geometries shift this transition to higher frequencies, thus achieving larger gain.

Based on the model findings, we develop optimized sensors to perform in vitro single-cell detection experiment. We find that both the PEDOT:PSS microelectrode and the OECT can monitor the cell adhesion process and recover their original performances after the cell detachment with trypsin. However, in the OECT the single-cell adhesion transient is measured with a current signal gain of $(20.2 \pm 0.9)$ dB at 625 Hz, in close agreement with the model prediction. Such an improvement is significant and facilitates measurement conditions regarding noise pickup and digitization, making OECT amplified impedance sensors a powerful tool for a new era of cell-substrate adhesion experiments with a single-cell resolution[5].

Importantly, our quantitative approach to compare bioelectronic impedance sensors is not limited to PEDOT:PSS based OECTs but can easily be extended to different electrolyte gated transistor architectures. The crucial parameter that describes the different material properties in our model is the channel transconductance. Large transconductances have been demonstrated for different channel materials and device architectures. They result from high capacitive couplings, as observed in organic mixed ionic and electronic conductors (e. g. PEDOT:PSS) or high carrier mobilities as found for example in graphene or carbon nanotubes. Current research in the material properties of such water stable semiconductors warrants further improvements in mobility or volumetric capacitance and will also augment the sensitivity of bioelectronic impedance sensors[42]. Our quantitative approach can serve as a guideline for the development of impedance sensors with maximized sensitivity and establishes a metric to compare future devices.

## Methods
### Device fabrication
Glass substrates ($50 \times 25$ mm$^2$) were cleaned by sonication in water and soap (10%)/acetone/isopropanol/distilled water baths. After a

dehydration step(10 min at 110 °C), the Microposit S1818 positive photoresist was spin coated (4000 rpm for 60 s) and annealed at 110 °C for 1 min. Metallic contacts were patterned through direct laser lithography by using the ML3 Microwriter (from Durham Magneto Optics). The photoresist was developed with Microposit MF-319 developer. Then, 10 nm of chromium and 25 nm of gold were deposited by thermal evaporation. Samples were immersed in acetone for 4 h for photoresist lift-off. Metallic contacts were encapsulated with the mr-DWL 5 negative photoresist (from Micro Resist Technology). The resin was spin coated at 3000 rpm for 30 s and annealed at 100 °C for 2 min. After laser exposure, samples were baked at 100 °C for 2 min and relaxed for 1 h at room temperature. Development was performed with mr-Dev 600 developer (Micro Resist Technology), and the resist was finally baked at 120 °C for 30 min. A double layer of S1818 was deposited and treated for 6 min in chlorobenzene for the photolithography of the PEDOT:PSS channel[54]. After the development, substrates were treated with air plasma (15 W for 2 min) and the PEDOT:PSS solution (94% PEDOT:PSS (Heraeus, Clevios PH1000), 5% of ethylene glycol (EG) (Sigma Aldrich), 1% of 3-glycidoxypropyltrimethoxysilane (GOPS), and 0.25% of 4-dodecylbenzenesulfonicacid (DBSA)) was spin coated at 3000 rpm for 10 s. The resulting film thickness was (100 ± 10) nm. The samples were subsequently annealed at 120 °C for 1 h, and S1818 was finally lifted-off after 4 h in acetone.

### Electrical measurements

DC characteristics of the OECTs were carried out with the Keysight 2912A source-measure unit (SMU), using a Ag/AgCl wire as gate electrode. The acquired data were analyzed to set the working point of the transistor during impedance sensing, i.e., the DC gate and drain voltages at which the transconductance $g_{m,DC}$ assumes its maximum value (Supplementary Fig. 2). AC measurements were performed with the MFLI lock-in amplifier (from Zurich Instruments). A constant DC offset voltage and a sinusoidal oscillation (amplitude 10 mV) with desired frequency were applied to the gate terminal. The resulting AC current flowing in the PEDOT:PSS channel was demodulated to acquire the current amplitude of the impedance sensors. In the OECT configuration, a constant DC voltage was applied between the source and the drain electrodes to bias the device at its working point. The microelectrode configuration was obtained by shorting the drain and the source terminals.

### Microparticle sensing experiment

To fabricate the AFM probe supporting the dielectric microparticle, we dispersed a powder of pyrophosphate microspheres (Polymat) on a glass slide. A PPP-NCHR cantilever of the Park NX10 AFM (force constant 34.55 N/m) was approached onto a drop of glue, and then put in contact with the upper part of a microsphere (with diameter of about 50 μm) until complete adhesion.

AFM measurements were performed in liquid, using 0.1 M PBS as electrolyte and an Ag/AgCl wire as gate electrode. To measure the current-distance spectroscopies, we placed the microparticle in the center of the PEDOT:PSS channel of the impedance sensor and we performed an AFM force-distance spectroscopy. The probe was lifted to the vertical coordinate $z_0$, distant 5 μm from the contact position, then gradually approached to the sample surface (scan speed 0.3 μm/s), and finally retracted again to $z_0$. During this process, we applied a constant modulation frequency to the gate terminal, and we measured the current amplitude flowing in the PEDOT:PSS layer, obtaining a current–distance spectroscopy curve. The experiment was repeated at 5 different frequencies (117, 330, 1170, 3330, and 11700 Hz) both in the OECT and in the microelectrode configurations.

### Single-cell detection experiment

The human malignant glioma cell line, T98G (CRL-1690™), derived from a glioblastoma multiform tumor, was selected for the single-cell detection experiment. This is characterized by indefinite lifespan and adherence properties previously reported also on PEDOT:PSS layers[55]. It was purchased from ATCC (Manassas, VA, USA) and cultured in Minimum Essential Medium (MEM) (Gibco™ 51200046, ThermoFisher scientific), supplemented with 10% fetal bovine serum, 1% L-glutamine, 10% sodium pyruvate and antibiotics (1% penicillin and 1% streptomycin) at 37 °C in 5% $CO_2$ incubator. All chemicals were from Merck. The experimental day, the sub-confluent (70–80%) cells population was detached by 0.25% trypsin in 0.02% EDTA (both from Merck) solution, re-suspended in fresh supplemented MEM and counted by the hemocytometric chamber. An aliquot, calculated to have a final density of $1 \times 10^3$ cells/cm³ was diluted in 600 μL of supplemented MEM and poured on the surface of the impedance sensors, previously sterilized by 20 min of UV exposure. A polydimethylsiloxane (PDMS) well was attached onto the sample substrate to host both the solution containing the cells and the Ag/AgCl gate electrode. After seeding, the cells reached the underlying substrate by gravity. For the experiment, we microfabricated a linear array of 10 PEDOT:PSS channels with dimensions $W \times L = 200 \times 50$ μm (an optical image is reported in Supplementary Fig. 6). In this way, the probability of a single cell falling onto a device channel after seeding was largely increased. Once this configuration was achieved, the sample was connected for the electrical measurements. The current spectra of the impedance sensor were acquired both in the OECT and the microelectrode configurations every 10 min after cell seeding. During each acquisition, the modulation frequency applied at the gate terminal was swept between 10 and $10^5$ Hz. At time $t = 200$ min, trypsin was used to completely remove the cells from the sample surface. Biological residuals were rinsed with PBS, and a final current spectrum was acquired in MEM to measure the sensor response after the cell detachment.

## Data availability

The data supporting the findings of this study are available within the article and its Supplementary Information as well as Source data. Source data are provided with this paper.

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

## Author contributions

T.C. conceived the experiments. F.B. and F.D. designed and fabricated the PEDOT:PSS impedance sensors. F.B. and T.C. performed the AFM experiments. I.Z. selected, cultured, and seeded the T98G cells for the in vitro experiment. F.B. and I.Z. carried out the single-cell sensing measurements. F.B. and T.C. wrote the manuscript with input from all other coauthors. B.F. and D.R. supervised the project.

## Competing interests

The authors declare no competing interests.
