## [Peer Review File · Nature Communications]

AC amplification gain in organic electrochemical transistors for impedance-based single cell sensorsREVIEWER COMMENTS

Reviewer #1 (Remarks to the Author):

The manuscript by Bonafè et al. is focused on the optimisation of OECT-based impedance sensor to record single cell adhesion and detachment transients. The article is overall clear and well written, the conclusion are supported by the data presented. However some major issues should be addressed before publication.

In the following the critical aspect are highlighted in a point by point fashion.

Issue#1: More emphasis should be provided on the importance of monitoring cell adhesion. The state of the art on cell adhesion monitoring should be discussed, pointing out the advantage of the approach herein reported.

Issue#2: Could the authors report in the introduction a more comprehensive description about ultra sensitive OECT biosensors? Have been ever employed to monitor single cell adhesion?

Issue#3: Is the Gate leakage correct of the OECT being monitored? Could the authors report the IG vg. Vg curves in Figure 1 along with the output and transfer characteristics of the transistor.

Issue #4: the authors state that "The proportionality constant p is independent on the sensor 174 configuration (OECT vs microelectrode) and we obtain its numerical value by fitting the microelectrode impedance spectra ". Could the authors comment in more detail the physical meaning of this parameter?

Issue #5: At page 8 the authors state "The factor f_{OECT} in eqn. 1 determines how the gate current is distributed between the source and the drain terminal and is typically assumed to be equal to 0.5 Several studies demonstrate that the f_{OECT} factor is slightly dependent on $V_{D,DC}$ and $215 V_{G,DC}$ as well as on channel geometry. For this reason, f_{OECT} was not assumed as constant in this experiment, but its value was set for each sensor in order to best fit the experimental data with the model." How the authors checked that this does not lead to overfitting? How this value has been selected. Did the author checked that the value assumed in each case are reasonably correct given the experimental conditions?

Issue #6: Figure 2 does not reports error bars at all. Hoe the repeatability and the reproducibility of the platform has been addressed?

Issue#7: Hoe the authors ensure that a single cell adhesion has been monitored? Did the monitor independently that a single cell was deposited on the channel of the OECT? could the authors comment more on this aspect?

Reviewer #2 (Remarks to the Author):

Bonafè and co-authors present a timely and thorough investigation of single-cell impedance sensing using electrolyte-gated transistor structures. I wholeheartedly agree with the authors' opinion that the field – in particular OECT and EGO-FET research – has focused more on functional demonstrations than on elucidating mechanisms of action. Given the ever-increasing literature utilizing these electrolyte-gated transistors for bioelectronics applications, I expect the contribution presented in this manuscript to be of great interest to a wide audience.

Overall, I liked the manuscript and feel that it should be accepted for publication after the authors address some relatively small issues, detailed below.

SCIENTIFIC ISSUES/QUESTIONS

1. Page 2, bottom: When discussing the nanophase network of organic semiconductors and polyelectrolytes, it would be good to mention the role of (primary, secondary) doping and additives (such as DMSO, PEG, etc).
2. I'm impressed by the sensitivities that the authors present, given the relatively minimal coverage of the active sensing area by the microparticle or the single cell. The authors do present a qualitative argument (page 6), but I'm interested in some mention of a quantitative argument as well. There is mention of the W/L ratio through the manuscript. Does the relative coverage of the W (or L) match expectations _quantitatively_? How is that the a nearly uncovered channel (e.g., Fig. 3a) exhibits such a strong change to the presence of the cell? (This might be covered in the referenced literature or distributed through the main and supplementary text, but it would be good to include this argument in the manuscript).
3. Following on this comment of cell coverage, I wonder if the authors investigated the OECT channels NOT covered 1 or more cells. Or even better, an OECT channel with a cell just off to the side. This could be a good control experiment to see if the T98G cells might be excreting adhesion substances that could alter the impedance.
4. Does the trypsin treatment (page 11) affect the OECT/microelectrode in any way? Relating to the previous comment, could it be "cleaning off" the channel of cell-excreted adhesion substances?
5. The derivation of Eqns. 3 and 4 is not directly obvious from "the two AC current contributions and differentiation". Since these equations are the real contribution of this manuscript, I would spend more space here elaborating the derivation so that all readers can clearly see the work.
6. Likewise, derivation of Eqn. 5 is not directly obvious. In addition, I would further expand Eqns. 3 and 4 until the frequency (ω) terms were visible. Since the remainder of the work focuses on frequency response, it would be very helpful to directly see s_{ch} and $s_{\mu E}$ as functions of ω .
7. In Figure 3b, the arrow indicating trypsin treatment appears at the timepoint after the drain current starts to increase again. Is this correct?

MINOR AND EDITORIAL ISSUES/QUESTIONS

8. Page 2, line 26: "mixed ionic and electronically" → "mixed ionic and electronic"
9. Page 3, line 2: "this properties combination" → "this combination"
10. Page 3, line 32: I would recommend defining the transfer function here, e.g., " I_{DS} as a function of V_G "
11. Figure 1b: Shouldn't the left and right contacts of the microelectrode be connected by a circuit line? On page 6, line 9, the authors state that they're short circuited.
12. Figure 1e: It would be nice to see the current amplitude as relative changes or normalized to initial value. This could help see the difference between OECT and microelectrode. Perhaps on the right axis?
13. Throughout: values do not need to be in parentheses. For example, "(0.059±0.002) nA/Ω" → "0.059±0.002 nA/Ω"
14. Page 7, line 1: "Figure 1e" → "Figure 1f"

15. Throughout: use “ \times ” instead of asterisk for multiplication. For example, “ $p*\partial d$ ” \rightarrow “ $p\times\partial d$ ”

Reviewer #3 (Remarks to the Author):

Bonafe et al. performed a quantitative study on the OECT amplification for cellular impedance sensing. This model related the sensor gain to OECT gm and device geometry. Instead of using cells, the authors used dielectric microparticles as their handling is easier. They used AFM cooperated impedance spectroscopy to measure the dielectric properties of these particles. They showed that the sensitivity is improved with transistor configuration compared to the microelectrode counterpart. The gain of the OECTs increased towards lower frequencies, as expected from OECTs and smaller geometries shifted this transition to higher frequencies due to their faster speed. As for novelty, impedance-based single-cell sensing using OECTs has been demonstrated before as well as the advantages of OECTs over electrodes for such biosensing applications. The impedance spectroscopy integrated AFM is known but applied here for the first time for oects to the best of my knowledge.

Other comments that should be addressed:

1. Single cell measurements were conducted by suspending 1000 of cells/cm³ on an OECT array assuming that a single cell would settle down on one channel. How do the authors confirm that what they detect is from a single cell?
2. Figure 1e, although the current amplitude increased with the OECT configuration, relative normalized responses calculated from $NR=(I-I_0)/I_0$ are identical for microelectrode and OECT configuration (0.2%). Then, how do the authors claim that the sensitivity improved for OECT configuration? The advantage of OECTs over microelectrodes should be the low noise level. However, the noise level of the current measured from both devices are also similar. What could be the reason?
3. How do the authors eliminate the effect of the cantilever on the current output during the electrical measurements?
4. How do the authors determine the optimum frequency (1.17 kHz)? The reason why 1.17 kHz was chosen should be explained before figure 1e and 1f.
5. “Qualitatively, this response is expected, as the microsphere represents a barrier for the ionic current in the electrolyte: when it is close to the sensor surface, the half-space through which ions can approach the active layer is reduced, thus increasing the effective impedance of the electrolyte Z_{el} . Consequently, upon approach, the interfacial impedance measured with the sensor increases and the AC current amplitude drops.” The reviewer does not agree with this statement. Z_{el} should not change for a fixed experimental system. It varies as a function of the distance between the electrodes, electrolyte conductivity, and the area of the electrodes. In addition to that, if we assume that the characteristic length, the distance between the electrode and OECT channel, is getting smaller and Z_{el} should decrease. Moreover, interfacial impedance should not change as a function of electrolyte resistance. It can only alter with charge and ion redistribution.
6. The proportionality constant (p) should be universal and does not change as a function of frequency or transistor or microelectrode parameters. Since the authors use different sizes of OECT, universality should be supported by experimental data.
7. “Accordingly, the sensitivity is given by the slope of the approach curves shown in Figure 1e.” this should be figure 1f?
8. The IV characteristics of three devices with different channel dimensions should be given in SI instead of normalized versions.
9. In figure 2d, why do the sensitivity curves (for 100x100 and 50x50) of two different channels overlap each other at high frequency? It conflicts with the explanation given below. “The plot of the current amplitude versus frequency for OECTs with different channel sizes clearly shows that with increasing channel area and length a strong reduction in f_c is observed.” “It is important to highlight that the position of $sOECT_{max}$ corresponds to the device cutoff frequency f_c (see Supp. Inf. S3 for the full mathematical treatment), and hence is a geometry-dependent

parameter." In the low-frequency regime, is the channel length dominant parameter and hence shows higher sensitivity while electrode area becomes more dominant at the high frequency?

10. Why was the 200x50 device geometry selected for the OECT and microelectrode comparison? How does the sensitivity change for the different dimensions of microelectrodes? Moreover, the ratio of sensitivities of OECT and microelectrode seems to be higher for 50x50 channel than other channels.

Why do authors move forward with 200x50 channel dimensions for the single cell measurements?

11. The common terminology is the electrolyte gated transistor, not water gated transistors although water can also be used as the dielectric (despite with lower efficiency).

Answer to Reviewers

Reviewer #1 (Remarks to the Author):

The manuscript by Bonafè et al. is focused on the optimization of OECT-based impedance sensor to record single cell adhesion and detachment transients. The article is overall clear and well written, the conclusions are supported by the data presented. However, some major issues should be addressed before publication. In the following the critical aspects are highlighted in a point-by-point fashion.

Issue#1: More emphasis should be provided on the importance of monitoring cell adhesion. The state of the art on cell adhesion monitoring should be discussed, pointing out the advantage of the approach herein reported.

Our reply: We agree with the Reviewer's comment, and we stress the importance of monitoring cellular adhesion in the reviewed version of the manuscript. Many studies demonstrate that cellular adhesion is a fundamental aspect in many biomedical processes such as wound healing, cancer development, and recognition processes in the immune system, and electrochemical impedance spectroscopy (EIS) is a well-established technique in this field of research. Anyway, the miniaturization of the sensing electrode leads to a dramatical increase in the impedance of noble metal-based microelectrodes, and experiments with single-cell resolution become very challenging with EIS. In our approach, we take advantage from both the large volumetric capacitance of PEDOT:PSS and the OECT amplification to provide a low-impedance and biocompatible platform able to detect single cell adhesion with a high signal-to-noise ratio.

Proposed change in the manuscript: We add the following section in the Introduction:

“Cell adhesion is an essential process in cell communication and regulation and becomes of fundamental importance in the development and maintenance of tissues.³⁰ Changes in cell adhesion can be the defining event in a wide range of diseases including arthritis,³¹ cancer,³² osteoporosis,³³ and atherosclerosis.³⁴ The study of single-cell adhesion is one of the most important and complicated aspects to understand in life sciences, with a considerable potential impact in bioelectronics. Over the years, numerous studies have shown the use of different techniques for the analysis of single-cell adhesion. Both the traction force microscopy (TFM)³⁶ and micropillar-array technique measure the cell adhesion force by monitoring the deformation induced on an elastic substrate.³⁷ Other methods include the use of Atomic Force Microscopy³⁸, optical tweezers³⁹ of microfluidics to assess the impact of cellular shape, size and deformability during adhesion⁴⁰. Despite their success in different demonstrations, these techniques rely on expensive equipment, are typically time consuming and

potentially can alter the cell behavior.⁴¹ An alternative non-invasive approach that combines scalability and real-time monitoring, is offered by electrical measurements that probe the electric cell-substrate impedance.⁴² In this technique, the cells are allowed to adhere directly onto the conductive surface of a functionalized sensor. A small AC voltage is applied and the ionic current that passes through the layer of adhering cells to the sensor is measured.⁴³ “

Issue#2: Could the authors report in the introduction a more comprehensive description about ultrasensitive OEET biosensors? Have been ever employed to monitor single cell adhesion?

Our reply: We follow the reviewer’s suggestion, and we add in the introduction of the revised manuscript a more comprehensive overview on ultrasensitive OEETs. Regarding cell adhesion, only a single experiment (Hemplel et. al Biosensors and Bioelectronics, Volume 180, 2021, 113101, <https://doi.org/10.1016/j.bios.2021.113101>) demonstrated the capability of OEET biosensors to monitor the presence of a single-cell on an active device. In this work, we extend this concept and demonstrate the fundamental device working principles. The outcome is a quantitative model of OEET amplification in AC measurements that defines the OEET gain and its relation to OEET device and material properties. With the findings we achieve monitoring of single cell adhesion transients.

Proposed change in the manuscript: We add the following section in the Introduction:

“Relying on the large transconductance combined with the biocompatible material properties, many research works propose OEETs as amplifying transistor to be integrated in electrochemical and bioelectronic sensors for healthcare applications.¹⁴ Established examples regard biosensors used to quantify the concentration of ionic or redox active analytes¹⁵. In this field, a further device functionalization with biorecognition elements has led to the ultrasensitive detection of RNA-biomarkers¹⁶, enzymes,¹⁷ and immunoglobulins with an unprecedented attomolar detection limit.¹⁸ Other successful applications of OEETs regard their use as potentiometric sensors for electrophysiological signals.¹⁹”

Issue#3: Is the Gate leakage correct of the OEET being monitored? Could the authors report the IG vs. V_g curves in Figure 1 along with the output and transfer characteristics of the transistor.

Our reply: We followed the suggestion of the reviewer and changed the figure. We note that the IG is too low to obtain quantitative information when plotted on the same scale as the drain current. For this reason, we add an additional plot showing only the leakage current of the OEET acquired during the DC characterization of the transistor in the Supporting Information S1. The ratio between the drain current flowing in the sensor and the leakage current remains below 0.3% during the DC characterization.

Proposed change in the manuscript: We plot the leakage currents acquired during the DC characterization of the transistors in the Supp. Inf. S1.

Issue #4: the authors state that "The proportionality constant p is independent on the sensor 174 configuration (OECT vs microelectrode) and we obtain its numerical value by fitting the microelectrode impedance spectra ". Could the authors comment in more detail the physical meaning of this parameter?

Our reply: The proportionality constant p quantifies how much the electrolyte resistance changes when the microparticle-channel distance d is modified (see Figure S2). Therefore, p is independent on the sensor configuration, but is only a measure of the variation of the electrolyte resistance when the geometry of the ionic current barrier (the microparticle) is modified. We add an additional sentence in the manuscript to clarify better the meaning of p .

Proposed change in the manuscript: We add a detailed insight on the extraction of the parameter p in the Supp. Inf. S5, and the following statement in the Results section:

"The proportionality constant p quantifies how much the electrolyte impedance changes when the microparticle-channel distance d is modified. Therefore, p is independent on the sensor configuration (OECT vs microelectrode) but is only a measure of the variation of the electrolyte impedance when the geometry of the ionic current barrier is modified."

Issue #5: At page 8 the authors state "The factor f_{OECT} in eqn. 1 determines how the gate current is distributed between the source and the drain terminal and is typically assumed to be equal to 0.5 Several studies demonstrate that the f_{OECT} factor is slightly dependent on $V_{D,DC}$ and $215 V_{G,DC}$ as well as on channel geometry. For this reason, f_{OECT} was not assumed as constant in this experiment, but its value was set for each sensor in order to best fit the experimental data with the model." How the authors checked that this does not lead to overfitting? How this value has been selected. Did the author checked that the value assumed in each case are reasonably correct given the experimental conditions?

Our reply: We thank the reviewer for this critical remark, and we investigated in more detail the value of f_{OECT} and its impact on the sensor performance. The findings are described in a new paragraph that is listed below. Briefly, we discard our initial hypothesis of $f_{OECT}=0.5$ that is typically assumed for DC measurement conditions. Instead, we found that in our AC measurement conditions, the impact of f_{OECT} simplifies and we can set its value to 1 for all cases covered in our work avoiding therefore any possible overfitting. The value $f_{OECT}=1$ implies that the AC gate current enters completely into the source electrode. The reason for this finding is that the gate current path into the

drain electrode is more resistive. We bias the drain at -0.4 V, therefore the channel region close to the drain electrode is depleted from hole carriers and much less conductive than the channel close to the source electrode.

The value of $f_{OECT}=1$ is confirmed by a solid fitting procedure that involves the frequency dependent data of the OECT sensitivity (Figure 2e). f_{OECT} impacts on the sensitivity at the high frequency limit. For small values of f_{OECT} , the source would exhibit only a small AC gate current contribution and hence at high frequencies, where the capacitive gate current dominates, the measured sensitivity would be very small as compared to the low frequency sensitivity. Instead, we observe that in the high frequency limit the OECT sensitivity is equal to the microelectrode sensitivity. Numerical fitting of this behavior leads to values of $f_{OECT} = 0.95 \pm 0.5$ for different geometries. Instead of using the numerical values we decided to simplify the argument in the manuscript and set the value to 1.0.

Proposed change in the manuscript: We include the previous discussion in the main text of the manuscript:

“The factor f_{OECT} in eqn. 1 determines how the gate current is distributed between the source and the drain terminal.⁴⁹ In general, the factor f_{OECT} depends on the bias conditions ($V_{D,DC}$ and $V_{G,DC}$), on channel geometry and on AC or DC measurement conditions.⁵⁰ In our case, we consider the AC transport regime where the gate current is a pure capacitive current without faradaic contributions. Further, in our biasing conditions ($V_{G,DC} = 0.1$ V and $V_{D,DC} = -0.4$ V) a significant negative potential is applied to the drain electrode leading to a depletion of holes from the channel region nearby the drain contact.¹⁷ As a result, the capacitive gate current encounters a resistive barrier at the drain electrode and instead enters into the source electrode. For this reason, we set $f_{OECT} = 1$ in our data analysis for each sensor geometry. The value is supported by numerical fitting procedures of our frequency dependent data leading to values close to one. (see Supp. Inf. S6). “

Issue #6: Figure 2 does not report error bars at all. How the repeatability and the reproducibility of the platform has been addressed?

Our reply: We reported the experimental error bars in Figure 2d/e/f as results from the averaging between the approach and lift curves of the AFM experiment. All plots show the transistor response on a logarithmic scale. On such a scale, the error assigned to experimental uncertainty becomes very small. We address the repeatability of the platform using the AFM to simulate the single cell detection in a first-order approximation. This allows to control the microparticle-channel distance with sub-micrometric precision, leading to highly repeatable experiments, as demonstrated by Figure 1e. In this way, we can provide a quantitative comparison between the microelectrode and the OECT

sensitivity which would be impossible through in-vitro experiments, due to (even small but significant) changes occurring during the adhesion of different cells. Regarding the reproducibility issue, all the device parameters used in the model are compatible with literature findings (see Table S5), and we used standard fabrication procedures (Decataldo et al. APL Materials 8, 091103 (2020) <https://doi.org/10.1063/5.0015232>) to realize our sensors. Thereby, given the wide literature regarding PEDOT:PSS-based OECTs, we expect that multiple repetitions of the experiment produce compatible results.

Proposed change in the manuscript: We present and discuss the full model parameters in Supp. Inf. S5.

Issue#7: How the authors ensure that a single cell adhesion has been monitored? Did they monitor independently that a single cell was deposited on the channel of the OECT? could the authors comment more on this aspect?

Our reply: We ensure the monitoring of a single cell adhesion with different combined strategies. Optical images showing a single T98G cell placed at the center of the PEDOT:PSS channel (see Figure 3 a) were acquired during the in-vitro experiment. A thick layer (5 μm) of negative photoresist insulated the metallic electrodes from all the remaining cells outlying the PEDOT:PSS active layer. The Ag/AgCl gate was placed in the reservoir above the sensors substrate to avoid its partial covering by cells, and its surface was optically monitored after cell seeding. We acquired in parallel the current spectrum of a control device (with the same dimensions and operating parameters) placed in the same reservoir, but with no cell seeded on the sensing channel, and we observed no significant alteration in its low-pass cutoff during the single-cell detection experiment (see Figure S10a). Finally, the recovery of the device original current spectrum after trypsinization demonstrated that our observations were not caused by other effects in the experimental setup (sensor degradation/contamination of the cell culture medium). The OECT DC transfer acquired after trypsinization (Figure S10b) and removal of the biological residuals with PBS demonstrates the correct working behavior of the OECT, indicating that the in-vitro experiment did not produce significant alteration in the sensor.

Proposed change in the manuscript: We extend the discussion on the control of the in vitro experiment in Supp. Inf. S10.

Reviewer #2 (Remarks to the Author):

Bonafè and co-authors present a timely and thorough investigation of single-cell impedance sensing using electrolyte-gated transistor structures. I wholeheartedly agree with the authors' opinion that the field – in particular OECT and EGOFET research – has focused more on functional demonstrations than on elucidating mechanisms of action. Given the ever-increasing literature utilizing these electrolyte-gated transistors for bioelectronics applications, I expect the contribution presented in this manuscript to be of great interest to a wide audience.

Overall, I liked the manuscript and feel that it should be accepted for publication after the authors address some relatively small issues, detailed below.

SCIENTIFIC ISSUES/QUESTIONS

1. Page 2, bottom: When discussing the nanophase network of organic semiconductors and polyelectrolytes, it would be good to mention the role of (primary, secondary) doping and additives (such as DMSO, PEG, etc).

Our reply: We mention the importance of additives and secondary dopants in the updated version of the manuscript to further complete our work. In particular, in the revised manuscript we highlight the importance of the addition of ethylene glycol (EG) to increase both PEDOT:PSS mobility and volumetric capacitance, as this process is a relevant step of our fabrication procedure.

Proposed change in the manuscript: We add the following statement in the manuscript introduction:

“The use of secondary dopant such as dimethyl sulfoxide (DMSO) and ethylene-glycol (EG) leads to further separation of PSS-rich islands from the conductive network of PEDOT, and therefore to a better conduction pathway and an increase of electrical conductivity.”

2. I'm impressed by the sensitivities that the authors present, given the relatively minimal coverage of the active sensing area by the microparticle or the single cell. The authors do present a qualitative argument (page 6), but I'm interested in some mention of a quantitative argument as well. There is mention of the W/L ratio through the manuscript. Does the relative coverage of the W (or L) match expectations _quantitatively_? How is that a nearly uncovered channel (e.g., Fig. 3a) exhibits such a strong change to the presence of the cell? (This might be covered in the referenced literature or distributed through the main and supplementary text, but it would be good to include this argument in the manuscript).

Our reply: Our data does not indicate that there is a simple linear relation between the coverage of the channel and the relative variation in OECT current response. For example, in S3 we provide data showing that the complete removal of the 50 um particle from the 100 x 100 um channel causes a relative change in current of ca. 5%. On the other hand, the cell with a cell body diameter of ca. 20um leads to a much larger variation exceeding 20%. The relatively small effect of the microparticle is explained by its spherical shape that creates only small direct contact area with the channel and allowing ions still to enter the OECT channel. Instead, to explain the unexpected large impact of the cell one needs to consider the biochemical processes underlying cellular adhesion. The T98G glioblastoma cells are an epithelial cell line that secrete a large amount of laminin and glycoproteins leading to the self-assembly of the basement membrane necessary for cell adhesion. (R. Kalluri “Basement Membranes: Structure, Assembly and Role in Tumor Angiogenesis Nat. Rev. 2003). The basement membrane is universally present in tissues of higher organisms as a dense, sheetlike structure with 50-100 nm thickness that is fundamental for the organization of epithelial tissue and creates a significant barrier to achieve division of tissue into compartments. Although our microscopy images cannot provide direct evidence for the formation of such a nanometric thick layer, we hypothesize that such a process is at the origin of the large impact of the cell’s adhesion on transistor response. It is likely that the secreted material acts like a strong barrier for ionic exchange between the OECT channel and the electrolyte and that the barrier extends below the cells body, covering larger parts of the OECT channel.

Proposed change in the manuscript: We add the following discussion in the main text of the manuscript:

“The frequency response of the OECT gain is well described by our model for both experiments, the single cell as well as the dielectric particle detection. However, the OECT current amplitude reduction is much larger for the case of the single cell even though the cell body has a diameter that is smaller than the dielectric particle. The effect is attributed to the much larger impedance increase caused by the cell adhering to the sensor surface. Glioblastoma tumor cells such as T98G secrete large amounts of laminin and glycoproteins to self-assemble the basement membrane below their cellular body.⁵³ We hypothesize that the basement membrane spreads below the cell body on top of the PEDOT:PSS channel and acts as a barrier increasing significantly the impedance. Trypsin treatment removes the cell body and dissolves also the basement membrane, making the effect reversible.”

3. Following on this comment of cell coverage, I wonder if the authors investigated the OECT channels NOT covered 1 or more cells. Or even better, an OECT channel with a cell just off to the side. This could be a good control experiment to see if the T98G cells might be excreting adhesion substances that could alter the impedance.

Our reply: During the single-cell detection experiment, we acquired in parallel the current spectrum of a control device (with the same dimensions and operating parameters) placed in the same reservoir, but with no cell seeded on the sensing channel, and we observed no significant alteration in its low-pass cutoff (see Figure S9a). This demonstrates that the variation of the sensor impedance spectrum is produced only by the adhesion of a single T98G cell.

4. Does the trypsin treatment (page 11) affect the OECT/microelectrode in any way? Relating to the previous comment, could it be “cleaning off” the channel of cell-excreted adhesion substances?

Our reply: After treatment with trypsin sensors recover their original current spectrum (see Figure 3b/c), and a final DC transfer acquired after the removal of the biological residuals with PBS demonstrates the correct working behavior of the OECT. As suggested by the Reviewer, this could demonstrate that trypsin “cleans off” the channel of cell-excreted adhesion substances.

Proposed change in the manuscript: We add the previous plot in the Supp. Inf. S10.

5. The derivation of Eqns. 3 and 4 is not directly obvious from “the two AC current contributions and differentiation”. Since these equations are the real contribution of this manuscript, I would spend more space here elaborating the derivation so that all readers can clearly see the work.

6. Likewise, derivation of Eqn. 5 is not directly obvious. In addition, I would further expand Eqns. 3 and 4 until the frequency (ω) terms were visible. Since the remainder of the work focuses on frequency response, it would be very helpful to directly see s_{ch} and $s_{\mu E}$ as functions of ω .

Our reply: We recognize the importance of eq. 3 and 4 as the main achievements of this work, but we prefer maintaining implicit mathematical expression to focus only on the basic line of argumentation with the equations. We provide the full mathematical derivation in the Supp. Inf.. Also we provide the explicit frequency dependence of the device sensitivities in a revised version of the Supp. Inf. (S7). We fully agree on the importance of Eqn.5 and we provide also its explicit dependence on frequency in the main manuscript.

7. In Figure 3b, the arrow indicating trypsin treatment appears at the timepoint after the drain current starts to increase again. Is this correct?

Our reply: Yes, after the complete adhesion of the cell on the PEDOT:PSS substrate the sensor response is slightly affected by some small fluctuations. Anyway, the recovery of the sensor current after treatment with trypsin is evident and starts from $t = 200$ min.

MINOR AND EDITORIAL ISSUES/QUESTIONS

8. Page 2, line 26: “mixed ionic and electronically” → “mixed ionic and electronic”

9. Page 3, line 2: “this properties combination” → “this combination”

10. Page 3, line 32: I would recommend defining the transfer function here, e.g., “ I_{DS} as a function of V_G ”

11. Figure 1b: Shouldn’t the left and right contacts of the microelectrode be connected by a circuit line? On page 6, line 9, the authors state that they’re short circuited.

12. Figure 1e: It would be nice to see the current amplitude as relative changes or normalized to initial value. This could help see the difference between OECT and microelectrode. Perhaps on the right axis?

13. Throughout: values do not need to be in parentheses. For example, “(0.059±0.002) nA/Ω” → 0.059±0.002 nA/Ω”

14. Page 7, line 1: “Figure 1e” → “Figure 1f”

15. Throughout: use “×” instead of asterisk for multiplication. For example, “p*∂d” → “p×∂d”

Proposed change in the manuscript: We modify all the editorial issues according to the Reviewer's statements, except from point 12. In this case, reporting normalized current values on the right axis would create some difficulties to read the graph. For completeness, we report normalized currents values in Supp. Inf. S4.

Reviewer #3 (Remarks to the Author):

Bonafè et al. performed a quantitative study on the OECT amplification for cellular impedance sensing. This model related the sensor gain to OECT gm and device geometry. Instead of using cells, the authors used dielectric microparticles as their handling is easier. They used AFM cooperated impedance spectroscopy to measure the dielectric properties of these particles. They showed that the sensitivity is improved with transistor configuration compared to the microelectrode counterpart. The gain of the OECTs increased towards lower frequencies, as expected from OECTs and smaller geometries shifted this transition to higher frequencies due to their faster speed. As for novelty, impedance-based single-cell sensing using OECTs has been demonstrated before as well as the advantages of OECTs over electrodes for such biosensing applications. The impedance spectroscopy integrated AFM is known but applied here for the first time for oects to the best of my knowledge.

Our reply: We thank the reviewer for the insightful and concise description of our work. At the same time, we would like to emphasize the elements of novelty introduced by this research. Scientific literature regarding OECTs as impedance biosensors has generally focused more on functional demonstrations than on elucidating mechanisms of action. In our work, we provide a quantitative comprehension of the gain effect for OECT-based impedance sensors which relates the device performance to its geometry and to fundamental material properties. We apply such a knowledge to a relevant field of biology (single cell sensing) in which OECTs have been successfully applied only in a single experiment (Hemplel et. al Biosensors and Bioelectronics, Volume 180, 2021, 113101, <https://doi.org/10.1016/j.bios.2021.113101>). We would also like to highlight the use of the AFM as a relevant element of novelty. The use of AFM mechanics allows to control the microparticle-channel distance with sub-micrometric precision, leading to highly repeatable experiments. Only in this way we can provide a quantitative comparison between the microelectrode and the OECT sensitivity, which would be impossible through in-vitro experiments, due to (even) small but significant changes occurring during the adhesion of different cells.

Other comments that should be addressed:

1. Single cell measurements were conducted by suspending 1000 of cells/cm³ on an OECT array assuming that a single cell would settle down on one channel. How do the authors confirm that what they detect is from a single cell?

Our reply: We ensure the monitoring of a single cell adhesion with different combined strategies. Optical images showing a single T98G cell placed at the center of the PEDOT:PSS channel (see Figure 3 a) were acquired during the in-vitro experiment. A thick layer (5 μ m) of negative photoresist insulated the metallic electrodes from all the remaining cells outlying the PEDOT:PSS active layer. The Ag/AgCl gate was placed in the reservoir above the sensors substrate to avoid its partial covering by cells, and its surface was optically monitored after cell seeding. We acquired in parallel the current spectrum of a control device (with the same dimensions and operating parameters) placed in the same reservoir, but with no cell seeded on the sensing channel, and we observed no significative alteration in its low-pass cutoff during the single-cell detection experiment (see Figure S10a). Finally, the recovery of the device original current spectrum after trypsinization demonstrated that our observations were not caused by other effects in the experimental setup (sensor degradation/contamination of the cell culture medium).

Proposed change in the manuscript: We add a further plot in Supp. Inf. S10, showing a final DC transfer acquired after trypsinization and remotion of the biological residuals. This furtherly demonstrates the correct working behavior of the OECT and thereby that our observations were not caused by material/device degradation.

2. Figure 1e, although the current amplitude increased with the OECT configuration, relative normalized responses calculated from $NR=(I-I_0)/I_0$ are identical for microelectrode and OECT configuration (0.2%). Then, how do the authors claim that the sensitivity improved for OECT configuration? The advantage of OECTs over microelectrodes should be the low noise level. However, the noise level of the current measured from both devices are also similar. What could be the reason?

Our reply: The reviewer's observations regarding the normalized response and the signal to noise ratio are correct. Even though the OECT introduces a significant gain that increases the measured signal level significantly, the relative noise level remains relatively constant in our experiment. To explain this finding, we have to consider different noise sources in microelectrode current recordings.

(Jia Yao and Kevin D. Gillis, “Quantification of Noise Sources for Amperometric Measurements ..”, Analyst 2012) We distinguish noise that is intrinsic to the electrode such as thermal current noise and shot noise from noise that is due to the recording electronics such as amplifier noise and input voltage noise. In our optimized microelectrode experiments the noise level is at ca. 80 pA at 1 kHz bandwidth and is mainly due to intrinsic thermal current noise (the current amplifier noise is specified as 1.4 pA at 1 kHz bandwidth – Femto DLPCA-200 amplifier at 10^7 V/A). Accordingly, the OECT amplifies the intrinsic noise and the signal to noise ratio remains constant in these experiments. Accordingly, the OECT cannot improve on the signal to noise ratio when the noise level is only determined by these intrinsic factors. The role of gain in improving signal to noise ratio becomes important when noise is introduced by the data acquisition system. In our case such noise is minimized in both cases (microelectrode and OECT) due to the use of very sophisticated signal conditioning circuits. For this reason, both signal traces in Figure 1f have a comparable signal to noise ratio. We note that in a realistic application scenario relying on a highly integrated array, microelectrode impedance recordings would be deteriorated due to a limited digital resolution.

Proposed change in the manuscript: We discuss the previous issue in Supp. Inf. S4, and we add a plot which compares the normalized currents measured with the OECT and the microelectrode devices.

3. How do the authors eliminate the effect of the cantilever on the current output during the electrical measurements?

Our reply: The presence of both the cantilever and the AFM stage changes the geometry of the liquid electrolyte and consequently modifies the electrolyte resistance R_{el} . Anyway, given the large diameter of the dielectric microparticle (50 μm) and the small displacement of the z-stage during the sensing experiment (5 μm), we expect that only the bottom part of the dielectric microparticle has an active role in modifying the ionic flow from the gate electrode to the sensing channel, simulating in first order approximation a biological cell which adheres to the sensor surface. A qualitative demonstration of this observation is furtherly provided in the reported experiment (left plot), where we measured the source current amplitude while gradually increasing the microparticle-channel distance d starting from the contact position. The microparticle displacement produces larger effect on the sensor response when the AFM probe is retracted for short distances from the contact position, indicating that the microparticle hindrance has a primary role in blocking the ionic flow from the electrolyte to the sensor channel. Such effect is furtherly highlighted by the plot on the right, where we report the OECT current variation per 1 μm -step as a function of d .

Proposed change in the manuscript: We add the previous discussion in Supp. Inf. S3.

4. How do the authors determine the optimum frequency (1.17 kHz)? The reason why 1.17 kHz was chosen should be explained before figure 1e and 1f.

Our reply: Experiments were performed at 5 fixed different frequencies (117, 330, 1170, 3330, and 11700 Hz) for each configuration (OECT/microelectrode) and for each channel geometry. The frequency 1.17 kHz does not correspond to an optimum frequency but is only reported as an example. We clarify this aspect in the reviewed version of the manuscript.

Proposed change in the manuscript: We specify that the 1.17 kHz frequency was only exemplificative:

“In Figure 1e we show the results of a typical microparticle distance - AC current experiment. conducted at 1.17 kHz excitation frequency, which was chosen as example.”

5. “Qualitatively, this response is expected, as the microsphere represents a barrier for the ionic current in the electrolyte: when it is close to the sensor surface, the half-space through which ions can approach the active layer is reduced, thus increasing the effective impedance of the electrolyte Z_{el} . Consequently, upon approach, the interfacial impedance measured with the sensor increases and the AC current amplitude drops.” The reviewer does not agree with this statement. Z_{el} should not change for a fixed experimental system. It varies as a function of the distance between the electrodes, electrolyte conductivity, and the area of the electrodes. In addition to that, if we assume that the characteristic length, the distance between the electrode and OECT channel, is getting smaller and Z_{el} should decrease. Moreover, interfacial impedance should not change as a function of electrolyte resistance. It can only alter with charge and ion redistribution.

Our reply: We agree with the reviewer's general arguments, and we fully confirm that a "Zel should not change for a fixed experimental system". We must remark though, that in our experiment, the geometry is not fixed. The dielectric microparticle is moved towards the surface and retracted. These motions are responsible for changes in impedance. All other aspects remain constant, as correctly requested by the reviewer (electrode geometry, area, ion concentration). We note that maybe the terminology Z_{el} is a bit misleading as it refers to a system that is composed of a conducting electrolyte and a dielectric microparticle. However, we prefer to keep this simplified notation and explain the situation more explicitly in the manuscript. Finally, when the reviewer writes "the distance between the electrode and OECT channel, is getting smaller and Zel should decrease", we want to stress that the distance between OECT channel and the reference electrode in our setup is not altered. Only the dielectric microparticle is moved. The particle is not conductive and not in contact to any external cable.

6. The proportionality constant (p) should be universal and does not change as a function of frequency or transistor or microelectrode parameters. Since the authors use different sizes of OECT, universality should be supported by experimental data.

Our reply: The parameter p is independent from the microelectrode/OECT configuration for each geometry (and of course from the frequency/device polarization) but resulted to be dependent on the channel geometry. We report the measured p parameters, and we discuss in detail their measurement in Supp. Inf. S5.

Proposed change in the manuscript: We discuss in detail the extraction of the parameter p and we provide the experimental results in Supp. Inf. S5.

7. "Accordingly, the sensitivity is given by the slope of the approach curves shown in Figure 1e." this should be figure 1f?

Our reply: Yes, we thank the Reviewer for the correction, and we modify the manuscript accordingly.

8. The IV characteristics of three devices with different channel dimensions should be given in SI instead of normalized versions.

Our reply: We provide the DC I-V characteristics of the OECT with different channel dimensions in the revised version of the Supp. Inf. S1.

9. In figure 2d, why do the sensitivity curves (for 100x100 and 50x50) of two different channels overlap each other at high frequency? It conflicts with the explanation given below.

“The plot of the current amplitude versus frequency for OECTs with different channel sizes clearly shows that with increasing channel area and length a strong reduction in f_c is observed.” “It is important to highlight that the position of $s_{OECTmax}$ corresponds to the device cutoff frequency f_c (see Supp. Inf. S3 for the full mathematical treatment), and hence is a geometry-dependent parameter.” In the low-frequency regime, is the channel length dominant parameter and hence shows higher sensitivity while electrode area becomes more dominant at the high frequency?

Our reply:

We thank the reviewer for these precise observations and his detailed interest into the predictions of our model. The interest prompted us to a revise the analysis of the experimental data and a consequent modification of the sensitivity plot. We report for completeness all the parameters used in the model in Supp. Inf. S6.

We agree with the reviewer’s observations, and we resume the main findings of the model with the following statements. (i) In the low frequency regime, the OECT sensitivity shows a linear increase with frequency until it reaches a sensitivity maximum $s_{OECTmax}$. In this frequency range the sensitivity increases strongly with the channel aspect ratio W/L as it is highly controlled by the OECT transconductance. (ii) In the intermediate regime, OECT impedance-based sensors have a maximum sensitivity at a defined operation frequency, which corresponds to their low-pass cutoff f_c . The spectral position of f_c (see eq. S8.1 in Supp. Inf.) is mainly determined by the channel area, which defines the channel capacitance C_{ch} and the electrolyte resistance R_{el} . Changes in the aspect ratio W/L modify the OECT transconductance and have a direct impact on the value of $s_{OECTmax}$. Accordingly, the frequency cutoffs of the 100x100 μm and the 200x50 μm structures are almost coincident, but the latter shows a higher sensitivity maximum. The smaller dimensions of the 50x50 μm device shift its f_c towards higher frequency, while maximum sensitivity is limited by the square aspect ratio. (iii) In

the high frequency limit the OEET sensitivity results from the capacitive gate current and is mostly controlled by the area of the channel.

We clarify these aspects in the revised manuscript according to the previous observations.

Proposed change in the manuscript: We add the previous discussion in the main text of the manuscript:

“Figure 2d shows the measured values for s_{OEET} obtained for three different channel geometries at different AC frequencies. Eq. 2 is in excellent agreement with the frequency dependence of the measured data, allowing to resume the main findings of the model with the following statements. (i) In the low frequency regime, the OEET sensitivity shows a linear increase with frequency until it reaches a sensitivity maximum $s_{OEETmax}$. In this frequency range the sensitivity increases strongly with the channel aspect ratio W/L as it is highly controlled by the OEET transconductance. (ii) In the intermediate regime, OEET impedance-based sensors have a maximum sensitivity at a defined operation frequency, which corresponds to their low-pass cutoff f_c . The spectral position of f_c (see eq. S8.1 in Supp. Inf.) is mainly determined by the channel area, which defines the channel capacitance C_{ch} and the electrolyte resistance R_{el} . Changes in the aspect ratio W/L modify the OEET transconductance and have a direct impact on the value of $s_{OEETmax}$. Accordingly, the frequency cutoffs of the $100 \times 100 \mu m$ and the $200 \times 50 \mu m$ structures are almost coincident, but the latter shows a higher sensitivity maximum. The smaller dimensions of the $50 \times 50 \mu m$ device shift its f_c towards higher frequency, while $s_{OEETmax}$ is limited by the square aspect ratio. (iii) In the high frequency limit the OEET sensitivity results from the capacitive gate current and is mostly controlled by the area of the channel.”

10. Why was the 200×50 device geometry selected for the OEET and microelectrode comparison? How does the sensitivity change for the different dimensions of microelectrodes? Moreover, the ratio of sensitivities of OEET and microelectrode seems to be higher for 50×50 channel than other channels. Why do authors move forward with 200×50 channel dimensions for the single cell measurements?

Our reply: The 200×50 device geometry was selected as exemplificative in Figure 2a to stress an important observation arising from our work: although the 50×50 geometry has the highest OEET gain, the 200×50 structure shows a higher absolute sensitivity (expressed in $\mu A/\Omega$), which is related to the device transconductance and channel size. The use of a 200×50 rectangular structure increases the device transconductance by a factor of 4 and reduces both the channel capacitive impedance and the electrolyte resistance. The combination of these factors shifts the low-pass cutoff towards smaller

frequencies, and lead to an increase of the maximum sensitivity (see eq. S3.2). For this reason, the 200x50 geometry was selected for the in-vitro experiment.

11. The common terminology is the electrolyte gated transistor, not water gated transistors although water can also be used as the dielectric (despite with lower efficiency).

Our reply: We agree that electrolyte gated transistor is a more general term. In our case the electrolyte is the NaCl contained in water. We chose “water gated” as it explains better why these devices are used in biosensors operating in contact with cells grown in aqueous solutions. However, we follow the reviewers suggestions and change the terminology.

REVIEWERS' COMMENTS

Reviewer #1 (Remarks to the Author):

The author have replied to all the issue risen by the Reviewer very carefully and accurately. However, the Reviewer still has some major concerns regarding the novelty of the work herein proposed and the interest for a broad audience. Indeed, the cell adhesions has been already successfully demonstrated by Hempel et. al using an OECT biosensors to monitor the presence of a single-cell on an active device. The present study represents an incremental research, and might be extremely interesting for a more specialist audience. The Reviewer, although has highly appreciated the work. by Bonafe et al., would therefore suggest a transfer to an other more focused journal.

Reviewer #2 (Remarks to the Author):

I feel that Bonafè et al more than satisfactorily addressed my concerns with the manuscript. I suggest that the revised manuscript be ACCEPTED for publication.

Reviewer #3 (Remarks to the Author):

The authors have responded to my questions.

Answer to Reviewers

Reviewer #1 (Remarks to the Author):

The authors have replied to all the issue risen by the Reviewer very carefully and accurately. However, the Reviewer still has some major concerns regarding the novelty of the work herein proposed and the interest for a broad audience. Indeed, the cell adhesions has been already successfully demonstrated by Hempel et. al using an OECT biosensors to monitor the presence of a single cell on an active device. The present study represents incremental research and might be extremely interesting for a more specialist audience. The Reviewer, although has highly appreciated the work. by Bonafe et al., would therefore suggest a transfer to another more focused journal.

Our reply: We see the novelty of our work in the model quantifying amplification in OECTs and the experimental demonstration of the model with a novel microscopy experiment. The single cell detection is described as an application of these findings. Our manuscript does not include the claim to provide single cell detection for the first time and we cite the related literature (for example Hempel et al.) in our manuscript. Accordingly, we agree with the reviewer on this, but we think that she/he did not fully appreciate the main implications of our work and the relevance of the quantitative approach. Until today, one finds many different biosensor articles that build on the hypothesis that transistor-based sensors intrinsically amplify measured signals. As our work demonstrates, amplification depends on many different parameters (transistor geometry, frequency, material properties etc.) and needs to be optimized. Our findings are valid for different electrolyte-based transistor architectures. For these reasons we see the broad relevance of our work that justifies publication in a journal targeting a non-specialized audience.